# DriveAction: A Benchmark for Exploring Human-like Driving Decisions in VLA Models

## Abstract

Vision-Language-Action (VLA) models have advanced autonomous driving, but existing benchmarks still lack scenario diversity, reliable action-level annotation, and evaluation protocols aligned with human preferences. To address these limitations, we introduce **DriveAction**, the first action-driven benchmark specifically designed for VLA models, comprising 16,185 QA pairs generated from 2,610 driving scenarios. DriveAction leverages real-world driving data proactively collected by drivers of autonomous vehicles to ensure broad and representative scenario coverage, offers high-level discrete action labels collected directly from drivers' actual driving operations, and implements an action-rooted tree-structured evaluation framework that explicitly links vision, language, and action tasks, supporting both comprehensive and task-specific assessment. Our experiments demonstrate that state-of-the-art vision-language models (VLMs) require both vision and language guidance for accurate action prediction: on average, accuracy drops by 3.3% without vision input, by 4.1% without language input, and by 8.0% without either. Our evaluation supports precise identification of model bottlenecks with robust and consistent results, thus providing new insights and a rigorous foundation for advancing human-like decisions in autonomous driving.

## 1 Introduction

Early autonomous driving systems were predominantly designed with a modular architecture (Huang et al., 2021; Wang et al., 2022; Shi et al., 2022; Huang et al., 2023; Shi et al., 2024), separating perception, prediction, planning, and control into independently optimized components. Recently, advances in large-scale multi-modal data and computational power have led to new paradigms, including end-to-end approaches (Hu et al., 2023; Jiang et al., 2023; Weng et al., 2024; Li et al., 2024; Tian et al., 2024) and Vision-Language-Action (VLA) models (Hwang et al., 2024; Xu et al., 2024; Zhou et al., 2025a;b), significantly enhancing system generalization and complex task performance. Despite these advancements, current systems still struggle with real-world diversity and replicating human driving preferences. Accordingly, within the VLA paradigm, comprehensive and rigorous evaluation of the entire pipeline is increasingly crucial in both academia and industry.

Recent benchmarks and datasets represent significant progress, yet they still struggle to capture the diversity, complexity, and behavioral characteristics of real-world driving. Most existing benchmarks (Qian et al., 2024; Guo et al., 2024; Nie et al., 2024; Sima et al., 2024) are constructed from open-source datasets (Dosovitskiy et al., 2017; Caesar et al., 2020; Sun et al., 2020; Mao et al., 2021; Krasin et al., 2017; Shang et al., 2019), resulting in limited source variety. Typically, these datasets are designed for object-level perception tasks and thus overlook the contextual richness and human intent intrinsic to realistic driving decisions. In addition, critical and challenging scenarios—such as road merges and exits, interactions with pedestrians, and construction zones—remain largely underrepresented, making evaluation results less relevant to practical deployment risks. Furthermore, the distribution of driver behaviors is highly imbalanced, with simple maneuvers like going straight dominating the data, while more complex events are insufficiently covered, leading to inadequate assessment of challenging behaviors.

Existing approaches to action ground truth annotation exhibit several limitations. Some works (Qian et al., 2024; Guo et al., 2024) do not provide action-level annotations and focus only on perception

or understanding tasks. Other approaches (Kim et al., 2018; Malla et al., 2023; Sima et al., 2024; Xie et al., 2025) utilize manually annotated action labels, but such labels are often generated after driving behavior occurs and therefore do not faithfully reflect real-time driving intent and decisions. This gap in high-fidelity action labels restricts the reliability and realism of current evaluation.

Regarding the design of evaluation systems, most existing benchmarks do not fully capture the core logic of driving decision-making. Some works focus primarily on isolated tasks such as video captioning (Kim et al., 2018), object recognition (Qian et al., 2024), or spatial understanding (Guo et al., 2024), without systematically covering the entire process from vision to action. For benchmarks that do address the full decision pipeline (Sima et al., 2024; Xie et al., 2025), a forward logic is often adopted, starting from perception and proceeding sequentially through prediction, planning, and action modules. However, this may not adequately represent a goal-driven paradigm that considers dependencies from the perspective of final decisions. As a result, current evaluation standards may not be closely aligned with realistic human driving decisions, highlighting the need for more comprehensive and goal-oriented evaluation frameworks.

To address these challenges, we present **DriveAction**, the first action-driven autonomous driving benchmark specifically designed for VLA models. Our key contributions are as follows:

- **Driver-Contributed Broad-Coverage Driving Scenarios.** DriveAction is constructed from real-world data proactively collected by drivers of autonomous vehicles, which fundamentally distinguishes it from existing benchmarks and provides a wide spectrum of both everyday and challenging driving scenarios. Manual curation guarantees a comprehensive and representative collection of driving scenarios and actions.

- **Human Driving Preference-Aligned Ground Truth.** Action labels are collected directly from real-time driving operations, faithfully capturing human intent at the moment of decision-making. To align with the output granularity of end-to-end large models, these labels are discretized into high-level actions, which reflect the categorical nature of human driving decisions. All labels are manually verified to ensure validity, with erroneous, unreasonable, or illegal behaviors excluded.

- **Action-Rooted Tree-Structured Evaluation.** DriveAction introduces an action-rooted, tree-structured evaluation framework, which dynamically determines the required vision and language tasks based on the target action and enables unified and systematic evaluation of the V-L-A pipeline. By supplying key scenario information, the framework enables evaluation within a realistic context and mitigates hallucinated outputs. It supports both comprehensive and task-specific evaluation, analyzing the effects of vision and language information on final action decisions and identifying model bottlenecks.

## 2 RELATED WORKS

### 2.1 AUTONOMOUS DRIVING MODELS

Autonomous driving was initially approached through modular systems (Huang et al., 2021; Wang et al., 2022; Shi et al., 2022; Huang et al., 2023; Shi et al., 2024), in which perception, prediction, planning, and control modules were developed and optimized independently. Subsequently, research progressed towards end-to-end architectures (Hu et al., 2023; Jiang et al., 2023; Weng et al., 2024; Li et al., 2024) that directly learn mappings from raw sensory inputs to actions. With advances in vision-language models (VLMs), hybrid approaches have emerged, in which VLMs are integrated into end-to-end architectures as independent modules that provide low-frequency driving suggestions (Tian et al., 2024). Most recently, the VLA paradigm (Hwang et al., 2024; Xu et al., 2024; Zhou et al., 2025a;b) has been established, enabling deeper integration of end-to-end models and VLMs for richer context understanding and greater generalization, further highlighting the growing need for dedicated and comprehensive evaluation benchmarks.

### 2.2 LANGUAGE-RELATED BENCHMARKS FOR AUTONOMOUS DRIVING

Existing works can be grouped into four categories. The first focuses on video captioning and explanation, represented by BDD-X (Kim et al., 2018), which links actions with textual descriptions, and DRAMA (Malla et al., 2023), which uses question-answer annotations to identify risk objects

and their corresponding causes. The second emphasizes spatial perception. For example, NuScenesQA (Qian et al., 2024) is designed for object-level and multi-modal question answering, while DriveMLLM (Guo et al., 2024) evaluates spatial and localization capabilities. The third focuses on general scene understanding, including TextVQA (Singh et al., 2019), Next-qa (Xiao et al., 2021), and RealWorldQA (xAI, 2024), which evaluate models on text recognition, video behavior comprehension, and visual understanding in real-world scenarios. The fourth emphasizes evaluation across the entire autonomous driving pipeline, including Reason2Drive (Nie et al., 2024), DriveLM (Sima et al., 2024), and DriveBench (Xie et al., 2025), which use a forward logic starting from perception, thus insufficiently aligned with goal-driven dependencies and realistic human driving behaviors.

Table 1: Comparison of DriveAction and Existing Benchmarks

| Benchmark | Scenario | Source | Label | Logic |
|---|---|---|---|---|
| BDD-X | Caption/explanation | Self-collected | Manual | None |
| DRAMA | Caption/explanation | Self-collected | Manual | Chain |
| NuScenesQA | spatial perception | nuScenes | None | None |
| DriveMLLM | spatial perception | nuScenes | None | None |
| TextVQA | general understanding | Open Images | None | None |
| Next-qa | general understanding | VidOR | None | None |
| RealWorldQA | general understanding | Self-collected | None | None |
| Reason2Drive | AD System | nuScenes, Waymo, ONCE | Open source | Chain |
| DriveLM | AD System | nuScenes, CARLA | Manual | Graph |
| DriveBench | AD System | DriveLM | Manual | Graph |
| **DriveAction** | AD System | Driver-contributed | Real-time operations | Tree |

## 3 DRIVEACTION

Inspired by existing benchmarks, we introduce DriveAction, the first action-driven benchmark specifically designed for VLA models, leveraging real-world driving preferences. The following sections provide a comprehensive description of DriveAction. Specifically, Section 3.1 highlights how driver-contributed and carefully curated data enable extensive scenario coverage and diverse action representation. Section 3.2 details action labels aligned with human-like decision making, collected from real-time driving operations and validated through multi-stage review. Section 3.3 introduces the action-rooted, tree-structured evaluation framework, supporting flexible and rigorous assessment of models across the full V-L-A pipeline.

### 3.1 DRIVER-CONTRIBUTED BROAD-COVERAGE DRIVING SCENARIOS

As shown in the *Source* column of Table 1, DriveAction uniquely aggregates real-world data collected by drivers of company-operated autonomous vehicles, in contrast to previous benchmarks that rely on self-collected or open-source data. Our dataset covers 148 cities and includes records of the complete lineup of mass-produced vehicles in our deployment. To ensure a comprehensive and representative collection of driving scenarios and actions, we perform multiple rounds of manual selection and quality control.

Table 2 summarizes the seven key scenario categories in DriveAction, each paired with representative actions and concise descriptions to illustrate the coverage and annotation strategy. These scenarios span a wide range of real-world driving conditions, including ramp and side road merging/splitting, as well as navigation- and efficiency-driven lane changes. The coverage extends beyond typical urban and highway environments to encompass challenging contexts such as complex intersections, construction zones, congestion, and interactions with vulnerable road users. Each scenario is associated with a variety of fine-grained actions—such as lane changes, deceleration, and bypass maneuvers—enabling detailed analysis of decision-making across diverse driving situations.

### 3.2 HUMAN DRIVING PREFERENCE-ALIGNED GROUND TRUTH

DriveAction derives action labels directly from actual driving operations, enabling accurate, real-time capture of driver intent and decisions. This contrasts with previous benchmarks, which rely on manual or open-source annotation, as indicated in the *Label* column of Table 1.

Table 2: Overview of Driving Scenarios and Action Categories in DriveAction

| Scenario | Actions | Description |
|---|---|---|
| On/Off Ramp | Forward-left, Forward-right | Merge/split at ramps |
| Main/Side Switch | Forward-left, Forward-right | Merge/split main and side roads |
| Navigation Lane Change | Lane change left/right | Change to the target lane as indicated by navigation |
| Efficiency Lane Change | Lane change left/right | Covers overtaking slow vehicles, avoiding stationary vehicles, and handling construction or congestion |
| Bypass VRU | Bypass left/right | For bypassing vulnerable road users |
| Intersection | Left, Right, Straight, U-turn, Stop, Decelerate, Forward/Backward left/right | Turning maneuvers at regular and complex intersections |
| Segment | Keep | Regular cruising scenarios on segments |

DriveAction adopts discretized high-level actions as ground truth, which matches the output granularity of end-to-end large models and better reflects the categorical nature of human decisions. For example, in lane change scenarios, instead of relying on trajectories sampled at high frequency, DriveAction captures decisions made at key points—such as whether to initiate a lane change and which direction to take—better matching the lower decision frequency of large models. This design provides a more suitable and fair evaluation standard for current autonomous driving models.

To ensure the reliability and validity of the action labels, all data undergo multiple rounds of manual verification, and instances with erroneous, unreasonable, or illegal behaviors are excluded. Specifically, this includes accidental control inputs such as mistaken acceleration or steering, behaviors inconsistent with the traffic environment such as abrupt stopping without obstacle, and violations of traffic regulations, including actions like crossing solid lane markings.

## 3.3 ACTION-ROOTED TREE-STRUCTURED EVALUATION

To establish an evaluation framework that systematically evaluates models across the full spectrum of real-world driving decisions, we propose an action-rooted, tree-structured evaluation architecture in DriveAction, which dynamically maps complex driving actions to the required vision and language tasks. The integration of rich contextual information ensures that decisions are made within a complete and realistic environment, and the highly flexible evaluation system supports both comprehensive V-L-A evaluation and task-specific evaluation.

### 3.3.1 TASK DEFINITION

Table 1 presents a comparison of the evaluation logic between DriveAction and existing benchmarks. Most current benchmarks either lack an explicit evaluation logic chain or, when such logic is present, adopt chain- or graph-based structures. DriveAction is the first to introduce an action-rooted, tree-structured framework as its evaluation logic. By leveraging action-driven tree dependencies, our evaluation paradigm systematically integrates V-L-A tasks into an extensible framework. This allows for dynamic subtask composition tailored to each action and enables comprehensive decision evaluation even in complex or long-tail scenarios. Such a structure substantially enhances both the expressiveness and future-oriented applicability of the benchmark.

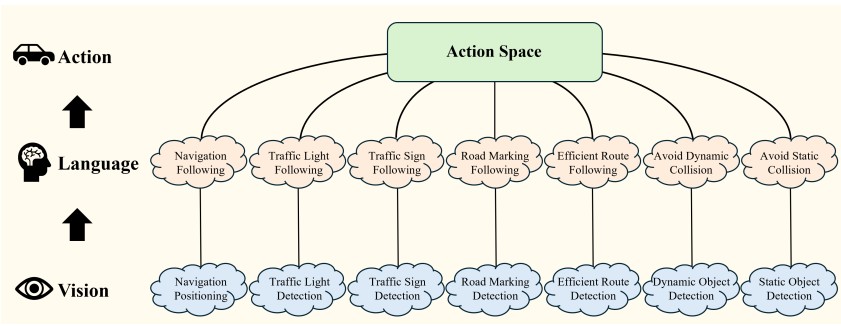

Figure 1: Action-Rooted Tree-Structured Task Architecture in DriveAction

Figure 1 illustrates the task architecture of DriveAction, which uses the action space as the root, explicitly defines the corresponding language tasks required for each action, and further maps each language task to its dependent vision tasks. This hierarchy is organized into three layers: the top layer consists of action nodes, such as lane change and intersection turning, representing final decision outputs of the model. The middle layer consists of language tasks, such as navigation following and traffic light following, which provide scene understanding for each action. The bottom layer comprises vision tasks, responsible for detecting and recognizing key environmental elements, such as lanes, traffic signs, and obstacles. Although the evaluation framework is modeled as a dependency tree descending from actions, the underlying model inference still follows the conventional V-L-A order. This design offers a systematic and targeted evaluation structure, thus closely matching the information flow and reasoning processes of real-world autonomous driving systems.

DriveAction comprises 14 independent tasks, including 7 vision tasks and 7 language tasks, as illustrated in Figure 1. All tasks are evaluated in a question-answering (QA) format, including both selection and judgment modes. Figure 2 presents the QA distribution across vision, language, and action levels, ensuring sufficient and representative coverage within each layer of the VLA hierarchy. For instance, turn actions constitute a significant portion, which aligns with their high frequency in real-world scenarios. All task definitions and example QA pairs are detailed in the Appendix.

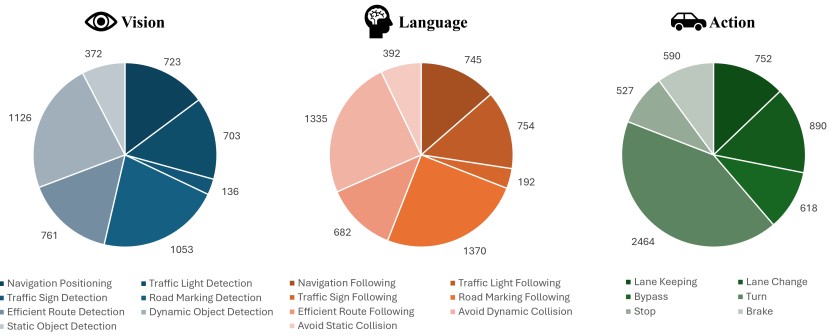

Figure 2: Distribution of QA Pairs Across Tasks in DriveAction

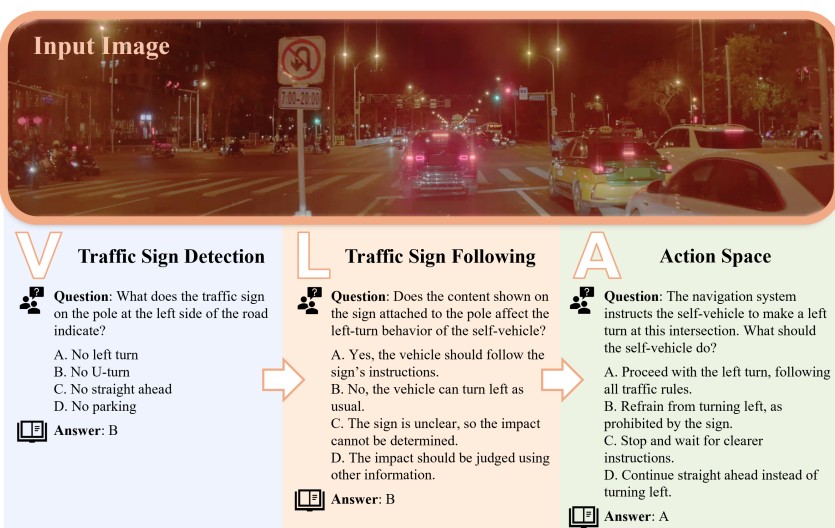

Figure 3: Example of the V-L-A Pipeline in Traffic Sign Task

Figure 3 uses the traffic sign task to exemplify the complete V-L-A evaluation pipeline in DriveAction. The process begins with detecting traffic signs, where the model is required to identify both their presence and type. Upon successful detection, the model must then interpret whether the sign is relevant to the current driving context and its own behavior. If the sign is deemed impactful,

the model is expected to make an appropriate driving decision accordingly. Through this V-L-A pipeline, DriveAction comprehensively evaluates the model's ability to transition from low-level perception to high-level understanding, and ultimately to informed decision-making.

### 3.3.2 SCENARIO INFORMATION DESIGN

DriveAction incorporates key scenario context into each evaluation prompt, ensuring that models are assessed under complete and realistic conditions, and mitigating hallucinatory reasoning. Specifically, three types of scenario information are provided:

- **Consecutive Visual Frames:** The model is given three consecutive visual frames captured immediately prior to the decision-making, thus supporting temporal reasoning in dynamic contexts.
- **Navigation Instruction:** Directly obtained from the in-vehicle navigation system, these instructions provide crucial route guidance, upcoming turns, and target lane information, thus defining clear decision objectives and path planning guidelines.
- **Vehicle Speeds:** Ego and target vehicle speeds, obtained from onboard sensors, quantify both the current and desired driving states. As dynamic information unavailable from images alone, they are essential for lane changes, overtaking, and acceleration decisions.

To assess the impact of scenario information, we examine the lane change task before an intersection under two conditions: with and without navigation instructions. As shown in Figure 4, the model reliably selects the correct lane when informed of an upcoming right turn, whereas lacking this guidance, it often makes "hallucinated" decisions misaligned with the driving goal. This highlights the necessity of scenario information for reliable and context-aware evaluation in autonomous driving.

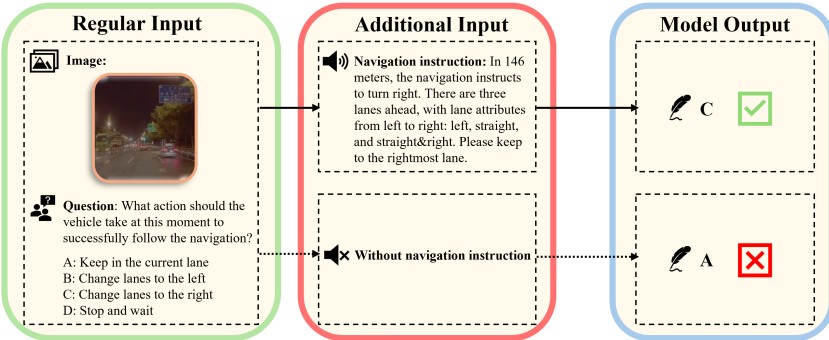

Figure 4: Effect of Navigation Information Design on Model Decision Evaluation

### 3.3.3 FLEXIBLE EVALUATION MODES

The architecture of VLA models requires evaluation methods that measure not only overall end-to-end performance, but also the effectiveness of each individual task. To this end, DriveAction introduces a flexible evaluation framework supporting both comprehensive and task-specific assessments. This enables analysis of how vision and language information influence action decisions and facilitates the identification of model bottlenecks within specific tasks.

The comprehensive evaluation focuses on the model's final decision outputs, where QA pairs from upstream vision (V) and language (L) tasks are optionally introduced as textual input for action decisions. Throughout all evaluations, basic scenario information is consistently provided. Four evaluation modes are supported:

- **Full Pipeline Mode (V-L-A):** Provides QA pairs from both V and L tasks, evaluating action performance under fully informed conditions.
- **Vision-Only Mode (V-A):** Only QA pairs from V tasks are provided, with no high-level L information available.
- **Language-Only Mode (L-A):** Only QA pairs from L tasks are provided, with no high-level V information available.

- **Uninformed Mode (A):** No upstream QA pairs are injected, evaluating the model's ability to make reasonable decisions relying purely on internal reasoning and existing knowledge without high-level external guidance.

By analyzing and comparing the results across these four modes, the framework can systematically reveal the model's reliance on different modalities and its generalization and reasoning abilities, thus supporting in-depth analysis of autonomous driving decision mechanisms.

The task-specific evaluation is conducted for each node in the hierarchical tree structure, providing fine-grained assessment of model capabilities. This approach yields valuable insights into the model's strengths and weaknesses in perception, reasoning, and decision-making skills, such as lane detection and traffic light recognition (vision tasks), navigation following and traffic rule understanding (language tasks), as well as concrete driving maneuvers such as lane changes and intersection turns (action tasks). While comprehensive evaluation measures overall decision-making, task-specific evaluation pinpoints the performance of individual components. Combining both offers a complete view of model capabilities.

## 4 EXPERIMENTS

In this section, we evaluate the performance of various VLMs on the DriveAction benchmark under multiple experimental settings. We present both comprehensive and task-specific evaluations to provide an in-depth assessment of model capabilities. Additionally, we include experiments with domain-specific driving models. Further experimental results are provided in the Appendix.

### 4.1 EXPERIMENTAL SETUP

We evaluate twelve widely adopted VLMs, divided into non-reasoning and reasoning categories. Non-reasoning models generate answers directly from input without explicit intermediate steps, and include GPT-4o (Hurst et al., 2024), GPT-4o mini (OpenAI, 2024a), GPT-4.1 (OpenAI, 2025a), GPT-4.1 mini (OpenAI, 2025a), Claude 3.5 Sonnet (Anthropic, 2024), Claude 3.7 Sonnet (Anthropic, 2025), and Qwen-Max-Latest (Bai et al., 2023). Reasoning models employ step-by-step reasoning to generate more human-like responses, and include o1 (OpenAI, 2024b), o3 (OpenAI, 2025b), Claude 3.7 Sonnet Thinking (Anthropic, 2025), Doubao-1.5-vision-pro-32k (Doubao, 2025), and Gemini 2.5 Pro (DeepMind, 2025). Model performance is measured by accuracy across all question types. All experiments are implemented using VLMEvalKit (Duan et al., 2024).

To provide further context within the driving domain, we conduct comparative evaluations against several benchmarks, including BDD-X (Kim et al., 2018), Next-qa (Xiao et al., 2021), TextVQA (Singh et al., 2019), RealWorldQA (xAI, 2024), and Reason2Drive (Nie et al., 2024). Since there are currently no open-sourced VLMs specifically finetuned for driving scenarios, we train two lightweight on-vehicle models with different architectures using proprietary driving data: one with a non-MOE architecture (0.5B) and the other with an MOE architecture (8×0.4B).

### 4.2 COMPREHENSIVE EVALUATION

Table 3: Model Performance (%) in Comprehensive Evaluation Modes

| Model | V-L-A | V-A | L-A | A |
|---|---|---|---|---|
| **Non-Reasoning** | | | | |
| GPT-4o | **88.84** | 84.72 | 86.52 | 81.01 |
| GPT-4o mini | **90.37** | 89.06 | 86.81 | 85.16 |
| GPT-4.1 | **90.35** | 85.95 | 87.53 | 81.71 |
| GPT-4.1 mini | **91.43** | 89.45 | 88.00 | 85.72 |
| Claude 3.5 Sonnet | **89.36** | 84.15 | 85.35 | 80.63 |
| Claude 3.7 Sonnet | **86.31** | 80.80 | 82.56 | 80.67 |
| Qwen-Max-Latest | **91.32** | 88.38 | 89.16 | 84.33 |
| **Reasoning** | | | | |
| o1 | **93.56** | 90.20 | 89.67 | 84.71 |
| o3 | **92.19** | 86.61 | 88.66 | 82.23 |
| Claude 3.7 Sonnet Thinking | **91.76** | 86.50 | 87.92 | 81.88 |
| Doubao-1.5-vision-pro-32k | **91.15** | 86.90 | 87.94 | 80.60 |
| Gemini 2.5 Pro | **91.86** | 86.81 | 88.93 | 83.60 |

Table 3 reports model accuracies under the four evaluation modes described in Section 3.3.3, investigating how access to visual and language information affects final action decisions in autonomous driving. Across the board, all models achieve their highest accuracy in the full pipeline mode (V-L-A) and the lowest in the uninformed mode (A). Notably, removing either visual or language modality leads to a consistent decline in performance. Our results reveal that state-of-the-art VLMs require both vision and language guidance for optimal decision making: on average, accuracy drops by 3.3% without vision input, by 4.1% without language input, and by 8.0% without either.

A closer comparison of different models reveals trends that generally align with their intended design. Reasoning models typically outperform non-reasoning models, especially in the V-L-A mode, where models such as o1 (OpenAI, 2024b) and o3 (OpenAI, 2025b) achieve the highest accuracies (exceeding 92%). However, this advantage does not always hold. In the A mode, some non-reasoning models perform as well as or better than reasoning models.

## 4.3 TASK-SPECIFIC EVALUATION

Table 4: Model Performance (%) on Navigation, Efficiency, Dynamic, and Static Tasks

| Model | Navigation | | | Efficiency | | | Dynamic | | | Static | | |
|---|---|---|---|---|---|---|---|---|---|---|---|---|
| | V | L | A | V | L | A | V | L | A | V | L | A |
| **Non-Reasoning** | | | | | | | | | | | | |
| GPT-4o | 66.8 | 75.2 | 78.2 | 73.7 | 84.1 | 54.8 | 87.3 | 93.8 | 98.9 | 87.0 | 97.2 | 93.3 |
| GPT-4o mini | 65.6 | 71.7 | 86.0 | 73.2 | 82.1 | 58.8 | 83.8 | 94.3 | 98.0 | 78.0 | 97.5 | 92.3 |
| GPT-4.1 | 71.3 | 77.7 | 82.7 | 82.8 | 85.6 | 61.1 | 89.5 | 96.7 | 99.4 | 85.9 | 99.0 | 91.3 |
| GPT-4.1 mini | 68.3 | 78.3 | 87.0 | 73.5 | 86.6 | 67.7 | 86.4 | 94.9 | 99.4 | 84.6 | 98.7 | 93.8 |
| Claude 3.5 Sonnet | 71.1 | 77.5 | 85.2 | 72.2 | 83.0 | 56.0 | 84.0 | 87.4 | 98.9 | 84.1 | 90.4 | 89.4 |
| Claude 3.7 Sonnet | 65.8 | 71.0 | 82.7 | 62.7 | 72.6 | 60.8 | 73.7 | 73.2 | 97.8 | 79.0 | 65.2 | 82.7 |
| Qwen-Max-Latest | 64.5 | 76.2 | 88.7 | 78.8 | 81.5 | 59.2 | 88.9 | 93.5 | 98.2 | 89.7 | 99.0 | 91.8 |
| **Reasoning** | | | | | | | | | | | | |
| o1 | 67.5 | 76.8 | 88.3 | 77.2 | 85.4 | 66.4 | 87.6 | 93.7 | 98.7 | 85.7 | 98.7 | 93.8 |
| o3 | 67.9 | 80.4 | 87.9 | 76.5 | 85.1 | 65.5 | 86.3 | 92.3 | 99.0 | 85.9 | 96.0 | 89.4 |
| Claude 3.7 Sonnet Thinking | 70.2 | 78.7 | 87.7 | 72.4 | 82.1 | 59.6 | 82.4 | 90.6 | 98.4 | 84.1 | 96.2 | 82.7 |
| Doubao-1.5-vision-pro-32k | 68.2 | 82.5 | 88.9 | 74.6 | 83.2 | 58.6 | 87.3 | 95.0 | 98.2 | 87.0 | 98.5 | 90.9 |
| Gemini 2.5 Pro | 70.9 | 79.6 | 89.7 | 75.9 | 85.3 | 71.1 | 89.6 | 92.6 | 99.5 | 85.1 | 97.7 | 90.4 |

Table 5: Model Performance (%) on Road Marking, Traffic Light, and Sign Tasks

| Model | Road Marking | | | Traffic Light | | | Sign | | |
|---|---|---|---|---|---|---|---|---|---|
| | V | L | A | V | L | A | V | L | A |
| **Non-Reasoning** | | | | | | | | | |
| GPT-4o | 76.4 | 90.4 | 94.0 | 56.7 | 82.2 | 65.7 | 77.9 | 80.3 | 82.0 |
| GPT-4o mini | 62.3 | 85.3 | 93.7 | 58.0 | 83.7 | 88.0 | 74.5 | 67.0 | 82.0 |
| GPT-4.1 | 73.1 | 91.7 | 93.1 | 67.2 | 82.9 | 61.4 | 83.2 | 87.1 | 83.1 |
| GPT-4.1 mini | 74.3 | 87.4 | 93.1 | 44.3 | 68.5 | 69.5 | 71.7 | 75.3 | 80.9 |
| Claude 3.5 Sonnet | 70.6 | 80.4 | 93.0 | 65.1 | 55.0 | 57.3 | 76.8 | 70.1 | 83.1 |
| Claude 3.7 Sonnet | 66.8 | 58.2 | 91.1 | 40.3 | 54.6 | 52.5 | 53.0 | 55.8 | 83.1 |
| Qwen-Max-Latest | 74.8 | 85.3 | 91.9 | 51.9 | 83.4 | 82.1 | 78.9 | 77.1 | 85.4 |
| **Reasoning** | | | | | | | | | |
| o1 | 73.8 | 92.0 | 92.6 | 59.3 | 84.3 | 68.3 | 72.3 | 81.3 | 84.3 |
| o3 | 70.9 | 83.5 | 89.9 | 49.8 | 61.7 | 54.8 | 72.3 | 77.8 | 79.8 |
| Claude 3.7 Sonnet Thinking | 68.7 | 87.4 | 90.5 | 54.0 | 70.0 | 60.0 | 68.5 | 70.9 | 82.0 |
| Doubao-1.5-vision-pro-32k | 82.6 | 88.0 | 91.5 | 56.6 | 75.4 | 59.7 | 82.3 | 81.2 | 84.3 |
| Gemini 2.5 Pro | 84.3 | 87.1 | 87.6 | 59.0 | 66.9 | 57.7 | 78.9 | 72.5 | 84.3 |

As summarized in Tables 4 and 5, we conduct a detailed task-specific evaluation to analyze model capabilities across diverse tasks, revealing substantial variability in performance across tasks and models. For example, in Table 4, models attain higher accuracy on Dynamic and Static tasks, which may be attributed to the prevalence and clear annotation of such cases in training data. The relatively strong performance on obstacle-related tasks compared to Efficiency tasks further suggests that current models adopt conservative strategies, favoring collision avoidance over optimizing for efficiency. In contrast, Navigation tasks remain a persistent challenge: while most models can respond to explicit navigation instructions, their substantially lower scores indicate limited proficiency in accurate lane localization and comprehensive navigation understanding.

As shown in Table 5, models again exhibit notable task-dependent variation across Road Marking, Traffic Light, and Sign tasks. Most models demonstrate strong performance on Road Marking and Sign tasks, whereas accuracy on Traffic Light tasks is consistently lower for several models, highlighting this area as a persistent bottleneck.

## 4.4 DRIVING DOMAIN MODELS EVALUATION

We evaluate domain-specific model performance on DriveAction and several existing benchmarks using the two proprietary on-vehicle models described above. To ensure fair comparison, the average score (0–100) across all questions is reported for each benchmark as a unified metric. As shown in Table 6, DriveAction reveals the most pronounced difference between models, while other benchmarks show only minor changes or decreases, indicating varying sensitivity to model improvements.

Table 6: Overall Performance of Driving Models Across Benchmarks

| Benchmark | Non-MOE (0.5B) | MOE (8×0.4B) | Score Difference |
|---|---|---|---|
| Reason2Drive | 78.87 | 67.75 | −11.12 |
| BDD-X | 52.81 | 50.11 | −2.70 |
| TextVQA | 39.04 | 41.30 | +2.26 |
| RealworldQA | 51.64 | 54.24 | +2.60 |
| NextQA | 55.02 | 59.23 | +4.21 |
| **DriveAction** | **67.40** | **79.78** | **+12.38** |

Tables 7 and 8 show task-specific results for the driving models. The MOE model consistently outperforms the non-MOE model on most tasks, especially in Navigation_L, Navigation_A, and Dynamic_L, with the exception being Traffic_Light_V. These gains are due to the MOE model's better interpretation of navigation instructions and dynamic risks, while the non-MOE model tends to default to 'keep lane' and underestimate obstacles. Importantly, despite the limited parameter scale, our on-vehicle models perform comparably to public generalist VLMs on most tasks.

Table 7: Driving Model Performance (%) in Task-Specific Evaluation (I)

| Model | Navigation | | | Efficiency | | | Dynamic | | | Static | | |
|---|---|---|---|---|---|---|---|---|---|---|---|---|
| | V | L | A | V | L | A | V | L | A | V | L | A |
| Non-MOE (0.5B) | 53.3 | 43.3 | 51.6 | 55.9 | 57.6 | 38.1 | 75.3 | 64.2 | 93.6 | 59.4 | 76.7 | 76.2 |
| MOE (8×0.4B) | 65.2 | 75.3 | 89.0 | 71.6 | 76.6 | 52.7 | 80.8 | 86.7 | 96.9 | 78.0 | 96.9 | 89.3 |
| Score Difference | +11.9 | +32.0 | +37.4 | +15.7 | +19.0 | +14.6 | +5.5 | +22.5 | +3.3 | +18.6 | +20.2 | +13.1 |

Table 8: Driving Model Performance (%) in Task-Specific Evaluation (II)

| Model | Road Marking | | | Traffic Light | | | Sign | | |
|---|---|---|---|---|---|---|---|---|---|
| | V | L | A | V | L | A | V | L | A |
| Non-MOE (0.5B) | 55.7 | 68.9 | 88.1 | 56.8 | 73.3 | 79.3 | 41.9 | 53.1 | 57.9 |
| MOE (8×0.4B) | 66.3 | 83.8 | 94.8 | 44.2 | 86.5 | 95.3 | 52.2 | 54.6 | 68.4 |
| Score Difference | +10.6 | +14.9 | +6.7 | -12.6 | +13.2 | +16.0 | +10.3 | +1.5 | +10.5 |

We further analyzed the unexpected performance drops on Reason2Drive (Nie et al., 2024) and BDD-X (Kim et al., 2018). For Reason2Drive (Nie et al., 2024), declines are mainly due to ambiguous or mislabeled questions about vehicle presence or position. For BDD-X (Kim et al., 2018), inconsistencies between scene descriptions and annotations in causal reasoning tasks, especially with traffic lights, negatively affect scores. These issues underscore limitations in question clarity and annotation quality in existing benchmarks.

## 5 CONCLUSION

In summary, we present DriveAction, the first action-driven benchmark for VLA models, introducing: (1) a broad-coverage driving dataset collected by drivers of autonomous vehicles; (2) human driving preference-aligned ground truth, with action labels from real-time driver operations and manual verification; and (3) an action-rooted, tree-structured evaluation framework for comprehensive, task-specific assessment. Our experiments demonstrate that model performance is highest when both visual and language inputs are provided, and drops when either modality is removed. DriveAction exhibits greater sensitivity and fine-grained discrimination between models compared to existing benchmarks. In future work, we aim to analyze model driving styles and support personalized model recommendations to better meet individual user needs.

ETHICS STATEMENT

All authors have carefully read and fully abide by the ICLR Code of Ethics. We confirm that this submission adheres to the highest standards of research integrity, fairness, and transparency.

Data Privacy and Ownership: The data used in this study were collected exclusively by company-operated vehicles and are solely owned by our company. No user or third-party data, personal information, or privacy-sensitive materials are involved, and the dataset does not contain any personally identifiable information from users. If externally captured images happen to contain personal information such as faces or license plates, these elements are strictly anonymized in accordance with all relevant regulatory requirements prior to any storage, processing, or analysis.

Human Annotation and Labor Practices: All human annotators who contributed to dataset labeling and verification were officially employed by our company through signed formal employment contracts. Their salaries were set above the legally mandated minimum wage in our region, and all annotators received benefits and protections in strict accordance with local labor laws and company regulations.

To the best of our knowledge, there are no conflicts of interest or other ethical issues associated with this work. We are committed to ongoing compliance with all relevant legal, institutional, and ethical guidelines as stipulated by the ICLR Code of Ethics.

REPRODUCIBILITY STATEMENT

We have made significant efforts to ensure the reproducibility of our benchmark. A detailed description of the data content and structure is provided in the supplementary materials, where a sample JSONL file is included. We will publicly release the entire dataset and related resources upon acceptance and the conclusion of the anonymity period, enabling researchers to fully reproduce and build upon our results.

LLM USAGE STATEMENT

LLMs were utilized in two aspects of this work. First, LLMs assisted in refining and optimizing the writing of the manuscript, including language polishing and structural adjustments. Second, LLMs played a role in generating QA pairs within the benchmark, following a controlled process and manual verification to ensure quality and accuracy. At all stages, the authors maintained full responsibility for the content and integrity of the work. All outputs from LLMs were thoroughly reviewed, curated, and validated by the authors.

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

## A  DATASET STRUCTURE AND ACCESS

This section presents a complete description of the dataset structure and all its fields:

- **question_slice_id**: The unique identifier for a slice. Multiple related questions may correspond to the same slice and share this ID.
- **qa_l0**: The primary task category of the question, corresponding to one of the three modalities: `Vision`, `Language`, or `Action`.
- **qa_l1**: The secondary task category of the question, specifying the concrete sub-task within the V-L-A categorization (e.g., `Navigation Position`).
- **question_category**: The type of question, which can be either `choice_questions` or `true_false_questions`.
- **content_cn / content_en**: The question and answer content in Chinese and English, respectively:
  - **question**: The question text.
  - **options** (only for `choice_questions`): All available options and their corresponding texts.
  - **answer**: The standard answer. For `choice_questions`, this is the option label; for `true_false_questions`, it is `True` or `False`.
- **image_0**, **image_1**, **image_2**: Three consecutive visual frames captured immediately prior to the decision-making moment, providing temporal context for dynamic scene understanding.

## B  TASK SPECIFICATIONS AND QA EXAMPLES

### B.1  DETAILED TASK SPECIFICATIONS

The benchmark includes the following tasks, each defined in the tree-structured task architecture introduced in the main text, with a specific evaluation focus as described below:

- **Navigation Position**: Concerns accurate localization of the ego vehicle within the current road structure, such as determining the exact lane and corresponding lane direction.
- **Navigation Following**: Focuses on understanding navigation instructions, including whether the vehicle is in a navigation-recommended lane and the potential for missed maneuvers due to lane changes or maintenance.
- **Efficient Route Detection**: Relates to the assessment of accessibility and suitability of the ego and adjacent lanes in terms of current traffic conditions.
- **Efficient Route Following**: Explores how lane accessibility and route conditions impact the ego vehicle's driving decisions and travel efficiency.
- **Dynamic Object Detection**: Deals with identifying the position, type, and motion of dynamic obstacles, such as vehicles or pedestrians, present in the scene.
- **Avoid Dynamic Collision**: Concerns the ability to identify whether dynamic obstacles pose a potential safety risk to the ego vehicle, with a focus on avoiding collisions.
- **Static Object Detection**: Addresses identification of the position and type of static obstacles, such as road facilities or fixed barriers.
- **Avoid Static Collision**: Relates to determining the potential safety risk posed by static obstacles, ensuring robust static collision avoidance.
- **Road Marking Detection**: Focuses on identifying road markings, including lane lines, crosswalks, stop lines, and directional arrows.
- **Road Marking Following**: Examines the understanding of road markings and their influence on the ego vehicle's driving behavior.
- **Traffic Light Detection**: Involves detecting the presence of traffic lights, as well as determining their number, types, and relevant directions in the scene.

- **Traffic Light Following**: Examines how recognized traffic light states affect the ego vehicle's passage through intersections, including appropriate stop or go actions.
- **Traffic Sign Detection**: Involves recognizing the contents of traffic signs and localizing their positions within the scene.
- **Traffic Sign Following**: Addresses whether recognized traffic sign information should influence the ego vehicle's driving behavior.

## B.2 REPRESENTATIVE QA EXAMPLES

All questions for each task are first generated as candidate QA pairs using a two-stage prompting framework with large language models (LLMs), and are subsequently screened by human annotators for validity. In the first stage, the LLM analyzes the scenario—leveraging key ground-truth attributes—to produce a structured scene report and assess whether specific environmental factors affect the ego vehicle's behavior. Based on this analysis, the second stage involves generating targeted, context-aware QA pairs. Table 9 presents a representative QA example for each task type, including a sample question, its corresponding answer, and an illustrative image.

Table 9: Representative QA examples for each task in the benchmark

| Task | Question | Answer | Image |
|------|----------|--------|-------|
| Navigation Position | 102 meters ahead, the navigation indicates a right turn. There are two lanes ahead, with the lane attributes from left to right being: straight + left turn, straight + right turn. Please drive in the rightmost lane. What is the attribute of the current lane?
A: Straight + Left Turn
B: Straight + Right Turn
C: Left Turn Only
D: Right Turn Only | A |  |
| Navigation Following | 104 meters ahead, the navigation indicates to proceed to the front right. There are four lanes ahead, with lane attributes from left to right being: bus lane, straight, straight, and right turn. Please drive in the rightmost lane. If your vehicle continues in the current lane, you will not be able to follow the navigation instructions smoothly. | True |  |
| Efficient Route Detection | Please judge based on the image: There is a large vehicle traveling in the lane to the right of your car, which may affect the space available for your car to change lanes to the right. | True |  |
| Efficient Route Following | As shown in the figure, the current speed of the vehicle is 8.61 km/h, while the ideal speed is 30.0 km/h. Based on the information in the image, which of the following best explains the difference between the current speed and the ideal speed?
A: There are no vehicles in front of the car, and the road is clear
B: The car is going downhill and needs to slow down
C: There are vehicles in front of the car, causing the actual speed to be lower than the ideal speed
D: The car is passing through a tunnel with a lower speed limit | C |  |

Table 9 – *Continued from previous page*

| Task | Question | Answer | Image |
|---|---|---|---|
| Dynamic Object Detection | Which of the following is a dynamic obstacle actually present in front of your vehicle during a right turn?
A: An electric two-wheeler crossing in front of your vehicle
B: A pedestrian walking on the crosswalk
C: A white van parked in the left lane
D: A taxi following behind your vehicle | A |  |
| Avoid Dynamic Collision | Based on the image, while making a left turn, which of the following is the most important potential dynamic safety risk factor at the current intersection that requires special attention?
A: Oncoming vehicles suddenly accelerating through the intersection
B: Motor vehicles near the zebra crossing on the right may cross into the vehicle's left-turn path
C: A truck parked on the left side of the road suddenly starting to move
D: Changes in the traffic light signal in the distance | B |  |
| Static Object Detection | Is there any static obstacle blocking the adjacent lane to the left of the current lane of your vehicle?
A: There is a guardrail
B: There is a flower bed
C: There is no static obstacle
D: There is a construction barrier | C |  |
| Avoid Static Collision | What impact do the presence of the curb and greenbelt on the right side of your vehicle have on making a right turn?
A: You can drive onto the sidewalk at will
B: You need to be careful to stay within the lane and avoid driving over the curb
C: You can cross the curb to make a right turn
D: You can temporarily park on the greenbelt | B |  |
| Road Marking Detection | As shown in the figure, what type of lane line is between the current lane and the adjacent lane on the left?
A: Solid line
B: Dashed line
C: Double solid line
D: No lane line | A |  |
| Road Marking Following | As shown in the figure, there is a central dividing line on the left side of the current lane. What impact does the presence of this dividing line have on the driving behavior of your vehicle?
A: You may change lanes across this dividing line at will
B: You may temporarily use the other lane to overtake
C: You may make a U-turn under any circumstances
D: You should stay within your own lane and must not cross this dividing line | D |  |

Table 9 – *Continued from previous page*

| Task | Question | Answer | Image |
|------|----------|--------|-------|
| Traffic Light Detection | The navigation indicates a right turn. As shown in the figure, please determine the type of traffic signal ahead in your current lane and the directions it controls. Choose the option that best matches the actual situation.
A: A left-turn arrow signal and a straight-through circular signal, controlling the left-turn and straight directions respectively
B: A single circular signal controlling all directions
C: Three circular signals, controlling left-turn, straight, and right-turn directions respectively
D: Only a straight-through circular signal, controlling both straight and right-turn directions | B | |
| Traffic Light Following | The navigation indicates a left turn. The traffic light shown in the picture indicates that the vehicle should turn left immediately. Is this statement correct? | False | |
| Traffic Sign Detection | In this scenario, what does the sign located on the right edge of the road remind vehicles of?
A: No entry
B: Construction
C: School zone
D: Detour indication | B | |
| Traffic Sign Following | When driving through a busy area, is it necessary to pay extra attention to pedestrians on the roadside?
A: Yes, because the sign warns to watch out for pedestrians
B: No, because there are no warning signs on this section
C: Only necessary when there are many vehicles
D: No special attention is needed because the speed is low | A | |

# C  EVALUATION AND CASE STUDY

## C.1  COMPARATIVE RESULTS AND ANALYSIS

Figure 5 presents a radar chart summarizing the decision-making performance of various VLMs across six distinct task categories: `Navigation`, `Efficiency`, `Dynamic`, `Static`, `Road Marking`, `Traffic Light`, and `Sign`. This visualization enables a direct comparison of model capabilities and highlights their strengths and weaknesses across different task categories.

As shown in Figure 5, VLMs achieve strong and consistent decision-making performance on the `Dynamic`, `Static`, and `Road Marking` categories, with scores that are both high and tightly clustered across different models, and most results in these first-tier categories are above 90%. Performance in the `Navigation` and `Sign` categories is slightly lower, generally ranging from 80% to 90%, and results remain stable and relatively concentrated, indicating that most VLMs handle these tasks with reasonable reliability.

In contrast, the `Efficiency` category demonstrates greater performance variability and generally lower scores, typically in the range of 50% to 70%, suggesting that these tasks are more difficult for existing VLMs. The `Traffic Light` category displays the largest spread among all models, with the highest and lowest scores differing by 35.5%, underscoring this as a particularly challenging area.

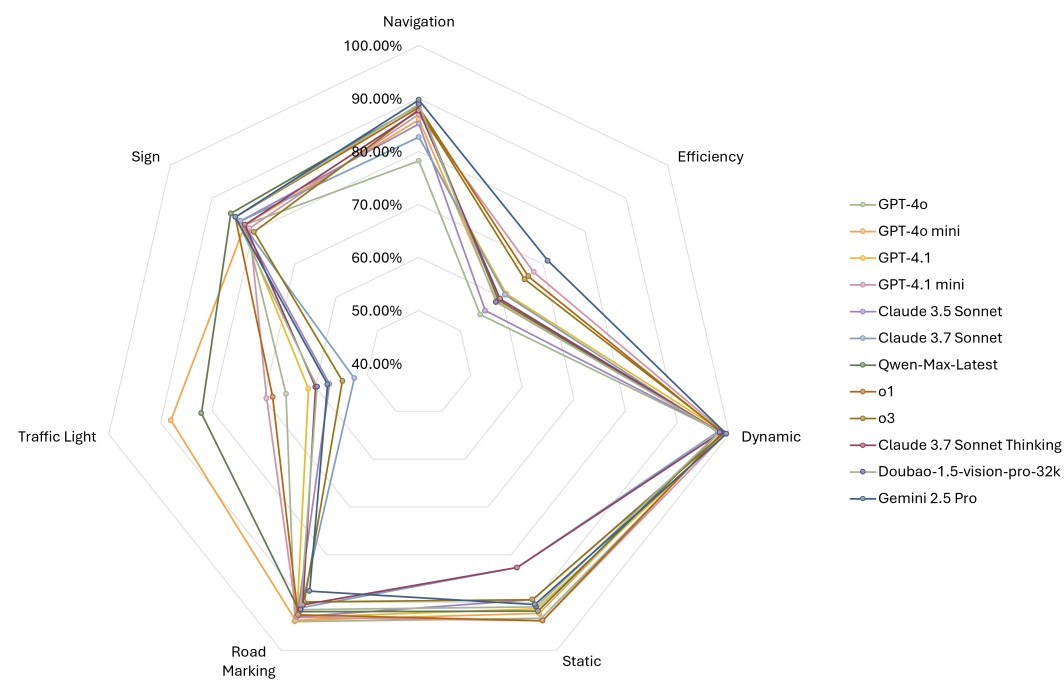

Figure 5: Model Performance (%) on Action Across Task Categories

## C.2 Comprehensive Evaluation Results: Case Study

To further investigate the impact of incorporating vision and language information on model performance, we conduct a case study focusing on GPT-4.1 (OpenAI, 2025a). As shown in comprehensive evaluation results, all models achieve their highest accuracy in the full pipeline mode (V-L-A) and the lowest in the uninformed mode (A). Based on this observation, we selected representative examples from various tasks where the model answered correctly under the V-L-A mode but failed with the A mode. These cases allow us to analyze in detail how the integration of vision and language information helps the model arrive at the correct answers and improve overall performance.

Table 10, Table 11, Table 12, and Table 13 present representative examples from the efficiency, navigation, traffic light, and road marking tasks, illustrating the impact of incorporating vision and language information on model performance. Each row in the table includes the input image (**Image**), input question (**Question**), relevant vision and language context (**V&L Info**), the ground-truth answer (**Ans**), the model's prediction under the A mode (**Pred-**), and the prediction under the V-L-A mode (**Pred+**).

The case study reveals several consistent patterns in model predictions when vision and language information are absent from the input. For efficiency and navigation tasks, the model tends to maintain the current lane by default, often overlooking construction zones, drivable areas, or the distinction between navigation and non-navigation lanes unless explicit visual or contextual cues are given. In the traffic light task, the model generally opts to stop and wait at intersections, showing limited ability to interpret the specific direction indicated by the light or discern its current signal state without additional information. For road marking scenarios, the model frequently chooses to change lanes in the navigation-indicated direction, neglecting lane markings or whether it is already in the correct lane; when obstacles are present, it is more likely to choose to slow down and wait, even in cases where overtaking would be permitted by the lane markings. These observations highlight the model's dependence on explicit vision and language signals to accurately understand and act in complex driving situations.

Table 10: Comprehensive Evaluation Examples on Efficiency Task (w/ & w/o V&L Info)

| Image | Question | V&L Info | Ans | Pred- | Pred+ |
|---|---|---|---|---|---|
|  | Based on the information in the image, what driving action and reasoning should the vehicle take in the current scenario? A: Continue straight, because the road ahead is clear B: Change to the left lane, because the left lane is wider C: Change to the right lane, because the right lane is unobstructed and passable D: Stop and wait, because there is a red light ahead | There are construction obstacles in front of the car. It is necessary to keep careful driving and proper distance when the vehicle is about to enter the ramp. | C | A | C |
|  | Based on the image, please determine which driving behavior the vehicle should adopt in the current scenario and explain the reason. A: Keep going straight, as there are no vehicles blocking ahead B: Change to the right lane, as the right lane is more open C: Change to the left lane, as the left lane is more open and helps improve traffic efficiency D: Stop immediately, as there is an obstacle ahead | There are vehicles in the left lane, but the distance is far and there is plenty of traffic space. There are vehicles in front of the car, and the left lane is relatively smooth. | C | A | C |
|  | As shown in the figure, the current speed of the vehicle is 80.78 km/h, and the ideal speed is 100.0 km/h. Given the current road conditions, which driving behavior should the driver adopt? A: Keep going straight, as the road ahead is clear B: Change lanes to the left to avoid the obstacle C: Change lanes to the right to avoid the construction area ahead D: Slow down and stop, waiting for the road ahead to clear | The front lane is closed by construction facilities and cannot continue to pass. The right lane is clear, providing safe lane-changing space for the self-vehicle. | C | B | C |

Table 11: Comprehensive Evaluation Examples on Navigation Task (w/ & w/o V&L Info)

| Image | Question | V&L Info | Ans | Pred- | Pred+ |
|---|---|---|---|---|---|
|  | 155 meters ahead, the navigation indicates a left turn. There are five lanes ahead, with the lane attributes from left to right being: left turn + U-turn, reversible lane, straight, straight, and bus lane. You are instructed to use the leftmost lane. What should you do now to successfully follow the navigation? A: Stay in the current lane B: Change to the right lane C: Change to the left lane D: Enter the bus lane | The allowed driving direction of the lane where the current vehicle is located is straight ahead. The vehicle's current lane is not a navigation lane. | C | A | C |

Table 11 – *Continued from previous page*

| Image | Question | V&L Info | Ans | Pred- | Pred+ |
|---|---|---|---|---|---|
|  | 83 meters ahead, the navigation indicates a right turn. There are two lanes ahead, with the lane attributes from left to right being: left turn, straight + right turn. Please use the rightmost lane. In the current scenario, what action should the vehicle take?
A: Stay in the current lane without changing direction
B: Change to the left lane to enter the left-turn lane
C: Change to the right lane to enter the straight + right-turn lane
D: Stop at the current position and do not choose any lane | The lane attribute of the current vehicle lane is left turn.

The current lane is not a navigation lane. | C | A | C |
|  | 83 meters ahead, the navigation indicates to proceed towards the right front. There are four lanes ahead, with lane attributes from left to right being: straight, straight, straight, right turn. You should use the rightmost lane. If you need to choose the navigation lane, in which direction should you change lanes?
A: Change lanes to the left
B: Stay in the current lane
C: Change lanes to the right
D: Stop and wait | The lane attribute of the current vehicle is straight ahead.

The current lane is not a navigation lane. | C | B | C |

Table 12: Comprehensive Evaluation Examples on Traffic Light Task (w/ & w/o V&L Info)

| Image | Question | V&L Info | Ans | Pred- | Pred+ |
|---|---|---|---|---|---|
|  | The navigation indicates a left turn. As shown in the picture, which driving action should the vehicle take?
A: Follow the left turn signal and immediately turn left through the intersection
B: Maintain current speed and go straight through the intersection
C: Stop and wait at the intersection
D: Immediately turn right onto the road on the right | The signal lights in front of the current self-lane include left-turn arrow lights and straight-ahead round lights.

At present, your own vehicle can't go straight through the intersection. | A | C | A |
|  | The navigation indicates a direction to the rear right, as shown in the figure. Considering the current traffic lights and the intersection situation, how should your vehicle proceed in compliance with the regulations?
A: Immediately turn left into the intersection
B: Immediately turn right into the intersection and pay attention to the situation at the intersection
C: Continue straight through the intersection
D: Stop and wait before the intersection | The type and control direction of the traffic light in front of the lane at the current intersection is a round light, which controls the straight direction.

At present, your own car can turn right directly through the intersection on the premise of ensuring safety. | B | D | B |

*Continued on next page*

Table 12 – *Continued from previous page*

| Image | Question | V&L Info | Ans | Pred- | Pred+ |
|---|---|---|---|---|---|
| 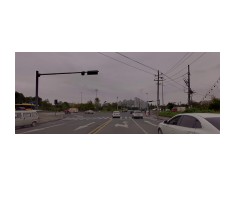 | The navigation indicates a left turn. As shown in the picture, how should your vehicle proceed in accordance with the regulations? A: Follow the left turn signal and immediately turn left to pass through the intersection B: Maintain your current speed and go straight through the intersection C: Immediately turn right onto the road on the right D: Stop and wait at the intersection | At present, the types and control directions of traffic lights in front of the self-lane are left turn+straight round lights and right turn arrow lights, which control the left-turn and straight-ahead directions respectively. At present, the traffic lights in front of the self-driving lane allow the self-driving vehicle to turn left. | A | D | A |

Table 13: Comprehensive Evaluation Examples on Road Marking Task (w/ & w/o V&L Info)

| Image | Question | V&L Info | Ans | Pred- | Pred+ |
|---|---|---|---|---|---|
| 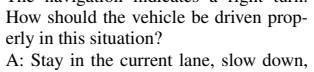 | The navigation indicates a left turn, as shown in the picture. How should the vehicle be driven properly in this situation? A: Change to the left lane and then turn left B: Turn left while staying in the current lane C: Change to the right lane and then turn left D: Make a U-turn | There is a left turn+straight arrow in the road sign at the intersection in front of the car. The road sign in front of the intersection can help the driver to judge whether the vehicle should continue driving in the current lane. | B | A | B |
| 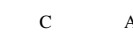 | The navigation indicates a right turn. How should the vehicle be driven properly in this situation? A: Stay in the current lane, slow down, observe the intersection, and then turn right B: Change to the left lane and then turn right C: Change to the right lane, enter the rightmost lane, and then turn right D: Go straight through the intersection | There are lane arrows and lane lines on the ground in front of the car. The road marking line between the right lane and the own lane means that the own vehicle cannot change lanes into the right lane at will. | A | C | A |
| 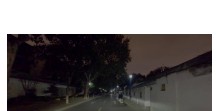 | As shown in the figure, the current speed of the vehicle is 25.71 km/h, and the ideal speed is 30.0 km/h. Given the current road conditions, what driving action should the vehicle take? A: Keep going straight, as the road ahead is clear B: Detour to the right, as there is more space on the right side C: Detour to the left, as there are no obstacles on the left and it can improve traffic efficiency D: Stop and wait for the obstacle ahead to move away | There are obstacles in front of the car, and the traffic on the right side is limited, but the traffic on the left side can pass. The current driving state of the vehicle is to keep driving in a straight line in the current lane. | C | D | C |

## C.3 TASK-SPECIFIC EVALUATION RESULTS: CASE STUDY

To gain deeper insights into the models' decision-making process across different driving scenarios, we conduct a case study based on task-specific evaluation using GPT-4.1 (OpenAI, 2025a). By analyzing representative examples from each task, we aim to understand the patterns and limitations in the model's predictions when handling real-world vision, language, and action challenges.

Table 14, Table 15, Table 16, Table 17, and Table 18 present representative examples from the navigation, efficiency, dynamic, road marking, and traffic light tasks respectively. Each row in the

table includes the input image (**Image**), the task type (**Type**),input question (**Question**), the ground-truth answer (**Ans**), and the model's prediction (**Pred**).

For the navigation task, we observe that while performance on the vision and language tasks is relatively lower, the accuracy in the action task remains high. As shown in Table 14, that even when the model is unable to accurately identify the vehicle's current lane, it can often infer the correct target lane by leveraging information from the navigation broadcast and the provided options.

Table 14: Task-specific Evaluation Examples on Navigation Task (Vision, Language, Action)

| Image | Type | Question | Ans | Pred |
|---|---|---|---|---|
| | Vision | 83 meters ahead, the navigation indicates a right turn. There are two lanes ahead, with lane attributes from left to right being: left turn, straight + right turn. Please use the rightmost lane. What is the lane attribute of your current lane?
A: Straight + Right Turn
B: Left Turn
C: Straight
D: Right Turn | B | A |
|  | Language | 83 meters ahead, the navigation indicates a right turn. There are two lanes ahead, with the lane attributes from left to right being: left turn, straight + right turn. Please use the rightmost lane. Is a right turn allowed from the rightmost lane?
A: Right turn is allowed
B: Only straight is allowed
C: Only left turn is allowed
D: Right turn is not allowed | A | A |
| | Action | 83 meters ahead, the navigation indicates a right turn. There are two lanes ahead, with the lane attributes from left to right being: left turn, straight + right turn. Please use the rightmost lane. In the current scenario, what action should the vehicle take?
A: Stay in the current lane without changing direction
B: Change to the left lane to enter the left-turn lane
C: Change to the right lane to enter the straight + right-turn lane
D: Stop at the current position and do not choose any lane | C | C |

In the efficiency task, we note that despite the model's strong performance on vision and language tasks, it may still fail to choose the correct action. Examples in Table 15 demonstrate that, although the model can correctly identify drivable areas that improve traffic flow, it sometimes conservatively opts to stay in the current lane rather than change lanes for greater efficiency.

Table 15: Task-specific Evaluation Examples on Efficiency Task (Vision, Language, Action)

| Image | Type | Question | Ans | Pred |
|---|---|---|---|---|
|  | Vision | Based on the image, assess the traffic conditions in the lane to the left of your vehicle. Which of the following descriptions best matches the current situation?
A: There is a vehicle in the left lane, but it is far away and there is ample space to pass
B: There are several vehicles closely lined up in the left lane, making it impossible to change lanes
C: The left lane is completely blocked by obstacles and cannot be used
D: There is a large truck very close to your vehicle in the left lane, making it impossible to change lanes | A | A |

*Continued on next page*

Table 15 – *Continued from previous page*

| Image | Type | Question | Ans | Pred |
|---|---|---|---|---|
| | Language | Please analyze the current driving environment of your vehicle based on the image information. Which of the following descriptions is the most accurate?
A: There are no vehicles in front of your car, and traffic is smooth
B: There is a vehicle blocking in front of your car, and the left lane is relatively clear
C: The right lane next to your car is completely empty, suitable for lane changing
D: Your car is surrounded by vehicles in front and behind, making lane changing impossible | B | B |
| | Action | Based on the image, please determine which driving behavior the vehicle should adopt in the current scenario and explain the reason
A: Keep going straight, as there are no vehicles blocking ahead
B: Change to the right lane, as the right lane is more open
C: Change to the left lane, as the left lane is more open and helps improve traffic efficiency
D: Stop immediately, as there is an obstacle ahead | C | A |

For the dynamic task, we find that action accuracy surpasses that of the vision and language tasks. As illustrated in Table 16, even if the model cannot precisely identify the most hazardous dynamic obstacles, it tends to select the safest options, such as braking or decelerating, based on empirical reasoning.

Table 16: Task-specific Evaluation Examples on Dynamic Task (Vision, Language, Action)

| Image | Type | Question | Ans | Pred |
|---|---|---|---|---|
| | Vision | Based on the image, what type of dynamic obstacles around the vehicle during a left turn may affect the vehicle's driving behavior?
A: There are several motor vehicles ahead making a left turn
B: There are pedestrians crossing the road ahead
C: There are non-motor vehicles going straight on the left side
D: There is an ambulance approaching from behind with its siren on | A | A |
|  | Language | Based on the image, when making a left turn at the current intersection, what is the most important potential dynamic safety risk factor to pay attention to?
A: A vehicle ahead turning left suddenly slows down or stops abruptly
B: An oncoming vehicle running a red light
C: The traffic light at the intersection suddenly turns red
D: The navigation system malfunctions | A | B |
| | Action | Based on the image, when making a left turn at the current intersection and facing a situation where there are many vehicles turning left ahead, what is the most standard driving behavior?
A: Maintain a low speed, slow down appropriately, and be ready to stop at any time
B: Accelerate to overtake the vehicles ahead
C: Change lanes frequently to look for gaps
D: Follow the vehicle in front closely and shorten the following distance | A | A |

In the road marking task, we observe significantly higher action scores compared to vision. Table 17 shows that, although the model may fail to recognize the presence of solid lane lines adjacent to the vehicle's current lane, it can still correctly choose to remain in the original lane. Similarly, even if the model does not detect the presence of a crosswalk, it is often able to infer from the options that it should slow down and yield to pedestrians.

Table 17: Task-specific Evaluation Examples on Road Marking Task (Vision, Language, Action)

| Image | Type | Question | Ans | Pred |
|---|---|---|---|---|
|  | Vision | As shown in the figure, what type of lane line is between the left side of your vehicle and the adjacent lane?
A: Solid line
B: Dashed line
C: Double yellow line
D: Curb | A | B |
| | Vision | As shown in the figure, is there a zebra crossing in front of the right-side sidewalk at the intersection ahead of your vehicle?
A: There is a zebra crossing
B: There is no zebra crossing
C: There is a left-turn waiting area
D: There is a stop line | A | B |
| | Action | As shown in the figure, the vehicle is driving normally in its current lane. Given the current road conditions, which of the following driving maneuvers is the most standard?
A: Change to the left lane
B: Change to the right lane
C: Continue driving in the current lane
D: Make a U-turn | C | C |
| | Action | The navigation indicates a right turn. How should the vehicle be driven in accordance with regulations?
A: Continue straight through the intersection at the current speed
B: Change lanes to the left lane
C: Slow down, observe if there are pedestrians at the crosswalk, and then turn right
D: Change lanes to the right and enter the non-motorized vehicle lane | C | C |

For the traffic light task, the model's action accuracy is notably low. Cases in Table 18 reveal that, even when the green light is present in the relevant direction, the model shows a tendency toward conservative behavior, such as waiting at the intersection rather than proceeding.

Table 18: Task-specific Evaluation Examples on Traffic Light Task (Vision, Language, Action)

| Image | Type | Question | Ans | Pred |
|---|---|---|---|---|
|  | Action | The navigation indicates a left turn. As shown in the figure, considering the current traffic signal and the navigation information, which of the following is the most standard driving behavior for your vehicle?
A: Comply with the left turn signal and make a left turn immediately
B: Maintain your current speed and go straight through the intersection
C: Make a right turn through the intersection immediately
D: Stop and wait, do not proceed through the intersection | A | D |

Table 18 – *Continued from previous page*

| Image | Type | Question | Ans | Pred |
|-------|------|----------|-----|------|
|  | Action | The navigation indicates to go straight at the intersection, as shown in the figure. Please choose the driving action that best fits the current scenario.
A: Proceed straight through the intersection at the current speed
B: Immediately turn left through the intersection
C: Immediately turn right through the intersection
D: Stop and wait before the intersection | A | D |

## C.4 STABILITY ANALYSIS

Table 19: Stability Analysis Under Different Modes

| Model | Mode | Run1 | Run2 | Run3 | Mean $\pm$ Std |
|-------|------|------|------|------|----------------|
| GPT-4.1 mini | V-L-A | 91.69 | 91.53 | 91.79 | 91.67 $\pm$ 0.13 |
|  | V-A | 89.06 | 89.59 | 89.23 | 89.29 $\pm$ 0.27 |
|  | L-A | 89.20 | 88.73 | 88.87 | 88.93 $\pm$ 0.24 |
|  | A | 84.71 | 84.50 | 84.78 | 84.66 $\pm$ 0.15 |
| Gemini 2.5 Pro | V-L-A | 91.25 | 91.22 | 91.43 | 91.30 $\pm$ 0.12 |
|  | V-A | 86.53 | 86.86 | 86.42 | 86.60 $\pm$ 0.23 |
|  | L-A | 88.99 | 88.58 | 88.66 | 88.75 $\pm$ 0.22 |
|  | A | 83.48 | 83.65 | 83.36 | 83.50 $\pm$ 0.14 |

To evaluate the consistency and robustness of model performance under different information input modes, we conduct a stability analysis by repeating each setting three times and reporting the mean and standard deviation for each model-mode pair, as summarized in Table 19. Across all settings, both GPT-4.1 mini (OpenAI, 2025a) and Gemini 2.5 Pro (DeepMind, 2025) demonstrate strong stability, with standard deviations generally below 0.3. These results indicate that our benchmark enables stable and objective evaluation of autonomous driving models, ensuring that performance measurements are reliable and reproducible across repeated trials.

