# OpenReview forum: "DriveAction: A Benchmark for Exploring Human-like Driving Decisions in VLA Models"
_ICLR.cc/2026/Conference — Submitted to ICLR 2026_

### Official Review · Reviewer_5d5K · 2025-10-27

**Soundness:** 3
**Presentation:** 3
**Contribution:** 2
**Rating:** 2
**Confidence:** 3

**Summary:**

This paper introduces DriveAction, a new VLA benchmark built from real-world driving data. The dataset contains QA pairs — including action questions — collected from deployed autonomous vehicles, and uses driver-labeled discrete action decisions. The benchmark covers a wide range of scenarios such as intersections, lane changes, and ramp merges that are useful for evaluating corner-case decision making. The authors propose an action-rooted evaluation framework and show results across various VLMs to demonstrate benchmark sensitivity.

**Strengths:**

The dataset comes from actual self-driving deployments, not synthetic or open-source simulator data. This includes real corner cases, high-value for both research and industry.

Human-validated action labels ensure the dataset has clean supervision and avoids spurious driving behaviors.

The scenarios selected are indeed realistic and relevant — these are good assets for the community to study.

**Weaknesses:**

I’m really unsure about the usefulness and motivation of this benchmark for VLA-based driving.
The actions are framed as multiple-choice. In real driving, there is no predefined set of choices popping up like a test. So I’m not convinced how this maps to actual autonomous driving:

Where would these choices come from at runtime?

Generated from another model? If so, that introduces another huge source of error.

Many examples feel tailor-crafted to match the scenario — unlikely to generalize.

Modern VLAs are being used to directly produce actions (i.e., trajectories or control tokens). In that case, direct prediction is simpler and more aligned with real-time operation than asking them to pick from provided abstract options.

To truly show real-world relevance, the authors need to demonstrate that better DriveAction performance → better driving.
For example:
1. improvements in collision rate
2. lower displacement error
3. higher success rate in closed-loop evaluation
None of that is measured here.

If the intention is instead to evaluate general reasoning, then the benchmark is too narrow — it only includes driving scenarios. In that framing, its impact would be limited because it can’t tell you anything about general VLM robustness across domains.

So either:
• The benchmark evaluates driving performance — then connect results to actual driving metrics.
• The benchmark evaluates reasoning — then it’s too domain-specific.
Right now it’s stuck in the middle — neither fully useful for driving systems nor for reasoning more broadly.

**Questions:**

Can the authors show that performance on DriveAction correlates with real driving outcomes (DE, CR, etc.)?

Can the authors justify how these multiple-choice actions would exist in an actual autonomous vehicle system?

---

> ### Author Response · Authors · 2025-12-04
>
> We agree that closed-loop trajectory evaluation and long-horizon decision consistency are crucial for assessing full VLA systems. However, this work targets a different but complementary problem: the lack of a systematic, action-rooted benchmark to evaluate VLMs themselves as potential foundations for VLA.
> ﻿
>
> Current end-to-end driving systems consider actions as the ultimate objective, but the community is increasingly exploring VLM-based approaches because of their strong reasoning ability and world knowledge, especially for handling long-tail cases. At the same time, as far as we know, no open-source VLM currently performs reliably enough in the driving domain to serve as a robust backbone for real-world VLA. Before closed-loop deployment, we need to clearly understand:
> ﻿
>
> - How well existing VLMs perform on driving-specific vision tasks (perception of lanes, signs, obstacles);
>
> - How well they handle scene-understanding language tasks (e.g., navigation following, traffic-light following);
>
> - How these capabilities translate into action decisions under an action-rooted decomposition.
>
>
> DriveAction is designed precisely for this purpose. It provides an open-loop, action-centered QA benchmark, where the V–L–A hierarchy is explicitly tied to target actions. This allows the community to:
> ﻿
>
> - Diagnose VLM capabilities and bottlenecks at each stage (vision, language, action) before integrating them into closed-loop controllers;
>
> - Compare different VLMs fairly and reproducibly on driving-relevant reasoning tasks;
>
> - Identify what is missing in current VLMs to make them viable foundations for future VLA systems.
>
>
> We will clarify in the paper that DriveAction is not meant to replace closed-loop trajectory evaluation, but to fill the current gap in action-rooted, VLM-centric benchmarking that is necessary to advance VLM-based autonomous driving research.

---

### Official Review · Reviewer_MFt4 · 2025-10-28

**Soundness:** 2
**Presentation:** 2
**Contribution:** 1
**Rating:** 2
**Confidence:** 3

**Summary:**

This paper introduces DriveAction, the first action-driven benchmark specifically designed for Vision-Language-Action (VLA) models in autonomous driving. DriveAction focuses on human-like decision-making. Extensive evaluations on 12 general VLMs and two domain-specific models (Non-MoE vs MoE) reveal how vision and language inputs affect final decisions and expose task-specific bottlenecks (e.g., navigation and traffic-light understanding).

**Strengths:**

1. Collecting 16k QA pairs from 2,610 real-world driving scenarios contributed by professional drivers.
2. Using real-time driver actions as ground-truth labels to capture authentic human decision intent.
3. Proposing an action-rooted tree-structured evaluation framework that connects vision, language, and action layers.

**Weaknesses:**

As we all know, VLA models are inherently action-centric, and thus the action dimension should play a more decisive role in evaluation. However, DriveAction primarily emphasizes open-loop QA assessments on Dynamic, Static, Navigation, and Efficiency tasks, rather than measuring closed-loop driving behavior that reflects real-time control and long-horizon decision consistency. So it makes me confused. I think the author needs to discuss more about the importance of this benchmark in the community.

**Questions:**

Same to Weaknesses.

---

> ### Author Response · Authors · 2025-12-04
>
> We agree that closed-loop trajectory evaluation and long-horizon decision consistency are crucial for assessing full VLA systems. However, this work targets a different but complementary problem: the lack of a systematic, action-rooted benchmark to evaluate VLMs themselves as potential foundations for VLA.
> ﻿
>
> Current end-to-end driving systems consider actions as the ultimate objective, but the community is increasingly exploring VLM-based approaches because of their strong reasoning ability and world knowledge, especially for handling long-tail cases. At the same time, as far as we know, no open-source VLM currently performs reliably enough in the driving domain to serve as a robust backbone for real-world VLA. Before closed-loop deployment, we need to clearly understand:
> ﻿
>
> - How well existing VLMs perform on driving-specific vision tasks (perception of lanes, signs, obstacles);
>
> - How well they handle scene-understanding language tasks (e.g., navigation following, traffic-light following);
>
> - How these capabilities translate into action decisions under an action-rooted decomposition.
>
>
> DriveAction is designed precisely for this purpose. It provides an open-loop, action-centered QA benchmark, where the V–L–A hierarchy is explicitly tied to target actions. This allows the community to:
> ﻿
>
> - Diagnose VLM capabilities and bottlenecks at each stage (vision, language, action) before integrating them into closed-loop controllers;
>
> - Compare different VLMs fairly and reproducibly on driving-relevant reasoning tasks;
>
> - Identify what is missing in current VLMs to make them viable foundations for future VLA systems.
>
>
> We will clarify in the paper that DriveAction is not meant to replace closed-loop trajectory evaluation, but to fill the current gap in action-rooted, VLM-centric benchmarking that is necessary to advance VLM-based autonomous driving research.

---

### Official Review · Reviewer_HC2D · 2025-11-02

**Soundness:** 3
**Presentation:** 3
**Contribution:** 2
**Rating:** 4
**Confidence:** 4

**Summary:**

The paper introduces DriveAction, a benchmark specifically designed for Vision-Language-Action (VLA) models in autonomous driving. It aims to fill gaps in scenario diversity, action-level annotation, and human-aligned evaluation. DriveAction includes 16,185 QA pairs across 2,610 driving scenarios, derived from driver-contributed real-world data, featuring action-rooted, tree-structured evaluation connecting vision, language, and action tasks. Experiments with 12 VLMs (e.g., GPT-4o, Claude 3.7, Gemini 2.5 Pro) reveal performance drops when either vision or language modalities are removed, highlighting multimodal dependence.

**Strengths:**

1. Proposes DriveAction, the first benchmark explicitly designed for Vision-Language-Action (VLA) evaluation in autonomous driving, addressing missing links between vision, language, and action reasoning.

2. Action labels are collected directly from real-time driver operations, faithfully capturing human decision intent rather than post-hoc annotations.

3. The action-rooted, tree-structured framework enables interpretable, modular analysis across V-L-A components, offering fine-grained evaluation flexibility.

4. Evaluates 12 state-of-the-art VLMs under four modality configurations (V-L-A / V-A / L-A / A), systematically showing modality dependencies.

**Weaknesses:**

1. While the benchmark is well-structured, its main finding (that models need both vision and language inputs) is intuitive and not conceptually groundbreaking.

2. Previous works like DriveLM (Sima et al., 2024) and Reason2Drive (Nie et al., 2024) already explore end-to-end reasoning or goal-driven evaluation, weakening the “first action-driven” claim.

3. Evaluation focuses on accuracy without deeper breakdowns (e.g., statistical variance, error typology, or causal reasoning analysis).

**Questions:**

1. How is inter-driver variability handled in “driver-contributed” data to ensure label consistency?

2. Could authors clarify whether the action labels are categorical only or also include continuous control values?

3. How are QA pairs validated for bias or ambiguity given LLM assistance in generation?

4. Do the results generalize to unseen city environments, or is there domain overfitting?

---

> ### Author Response · Authors · 2025-12-04
>
> ## Response to Weakness 1
>
> As defined in our paper, the V–L–A hierarchy in DriveAction refers to:
> ﻿
>
> **V (vision tasks)**: perception-level QA (lanes, traffic signs, obstacles, etc.);
>
> **L (language tasks)**: scene-understanding QA (e.g., navigation following, traffic light following) that encode high-level semantics;
>
> **A (actions)**: final decision nodes (e.g., lane change, intersection turning).
>
>
> What our benchmark investigates is how upstream vision (V) and language (L) QA performance quantitatively relates to downstream action (A) accuracy. Our benchmark can precisely distinguish model strengths/weaknesses at each V–L–A level, revealing concrete bottlenecks. We will make this distinction clearer in the revision.
>
>
> ## Response to Weakness 2
>
> We will clarify what we mean by “action-driven”.
> ﻿
>
> Unlike prior works that mainly provide free-form reasoning evaluations, DriveAction is action-rooted and tree-structured, which dynamically determines the required vision and language tasks based on the target action and enables unified and systematic evaluation of the V-L-A pipeline.
>
> ## Response to Weakness 3
>
> Beyond overall accuracy, our evaluation already includes more detailed analyses in **Appendix C** (Evaluation and Case Study), including comparative results across V–L–A levels and case studies. We will make these components more explicit in the main text by clearly referencing Appendix C and briefly summarizing the key breakdown results.

---

### Official Review · Reviewer_KDFH · 2025-11-04

**Soundness:** 3
**Presentation:** 3
**Contribution:** 2
**Rating:** 4
**Confidence:** 4

**Summary:**

This paper introduces DriveAction, an action-driven benchmark for Vision-Language-Action (VLA) models in autonomous driving. It includes over 16K QA pairs across diverse real-world scenarios with human-annotated action labels and a tree-structured evaluation framework, enabling comprehensive assessment of vision, language, and action reasoning.

**Strengths:**

1. The paper introduces DriveAction, a well-structured benchmark.

2. Dataset quality is high, with real-world, driver-contributed data with diverse scenarios.

**Weaknesses:**

1. I suggest that the authors include a discussion of recent studies on VLM-generated datasets for autonomous driving that are built on different foundations. For example, some works such as [1][2] generate data based on existing datasets like nuScenes or nuPlan, while others use internal datasets. Highlighting these distinctions would help the community better understand the overall differences and positioning of this work.

[1] Y. Xu et al., “VLM-AD: End-to-End Autonomous Driving through Vision-Language Model Supervision,” CoRL, 2025

[2] Z. Zhou et al., “AutoVLA: A Vision-Language-Action Model for End-to-End Autonomous Driving with Adaptive Reasoning and Reinforcement Fine-Tuning,” NeurIPS, 2025.

2. Another concern is dataset quality. Although human verification is mentioned, the annotation and quality-control process could be described in greater detail to improve transparency and reproducibility.

3. While the benchmark design is strong, the paper mainly focuses on dataset construction and evaluation, with limited methodological novelty. I am not sure whether it fits better under a benchmark or dataset track, if such a category exists.

**Questions:**

1. How is DriveAction’s action taxonomy defined and maintained to prevent overlap or ambiguity between tasks (e.g., “navigation lane change” vs. “efficiency lane change”)?


2. Were there any efforts to balance action categories, given the natural bias toward simple maneuvers (e.g., going straight)?

---

> ### Author Response · Authors · 2025-12-04
>
> ## Response to Weakness 1
>
> We thank the reviewer for pointing out these important recent works. We have carefully studied VLM-AD [1] and AutoVLA [2] and will add a dedicated discussion in the related work section. Our positioning is as follows:
> ﻿
>
> **Action labels source and granularity.** In our dataset, action labels are collected directly from real-time human driving operations, capturing the driver’s intent at the exact decision moment. Moreover, the action taxonomy is highly fine-grained and scenario-aware, including categories such as On/Off Ramp, Main/Side Switch, Navigation Lane Change, Efficiency Lane Change, Bypass VRU, Intersection, Segment, etc. This allows us to evaluate V-L-A models in specific, complex real-world scenarios and under explicit traffic rules. In contrast, existing works adopt more generic action spaces (e.g., VLM-AD’s control/turn/lane action lists), which make it difficult to diagnose model behavior in particular sub-scenarios. To the best of our knowledge, our level of scenario-specific action refinement is not yet present in prior VLM-generated driving datasets.
> ﻿
>
> **CoT generation.** Both VLM-AD and AutoVLA leverage general-purpose open-source VLMs to generate driving Chain-of-Thought (CoT) explanations. Given the still-limited reliability of current VLMs in the driving domain, the quality and relevance of such free-form reasoning can be hard to guarantee. In contrast, our dataset is built upon an action-rooted, tree-structured evaluation framework, which dynamically determines the required vision and language tasks conditioned on the target action. This structure explicitly guides the annotation to focus on the key factors influencing the action and suppresses irrelevant or spurious reasoning, leading to more systematic and interpretable supervision. We will highlight these distinctions and better clarify the positioning of our work relative to [1,2] in the revised manuscript.
>
>
> ## Response to Weakness 2
>
>
> We agree that more detail on annotation and quality control will improve transparency. In the revision, we will expand the description as follows:
> ﻿
>
> Multi-stage human verification. After collecting action labels from real driving logs, each sample is first automatically filtered by basic consistency checks (e.g., speed vs. braking, lane position vs. lane-change labels). Then, trained annotators perform multiple rounds of manual review, checking the video, sensor context, and control signals together.
> ﻿
>
> Explicit exclusion criteria. During manual verification, annotators remove samples that (1) clearly result from accidental control inputs (e.g., unintended acceleration/steering), (2) are inconsistent with the visible traffic environment (e.g., abrupt stopping without any obstacle or rule-based reason), or (3) violate traffic regulations (e.g., crossing solid lane markings, illegal lane changes, ignoring red lights). Only samples passing all criteria are retained.
> ﻿
>
> We believe these additions address the reviewer’s concern and make the dataset construction process more transparent and reproducible.
>
>
> ## Response to Weakness 3
>
>
> Our submission is indeed primarily a dataset and benchmark paper, and we have explicitly selected “Datasets and Benchmarks” as the primary area at submission time to align with the intended contribution type.

---

> ### Author Response · Authors · 2025-12-04
>
> ## Response to Question 1
>
>
> Our action taxonomy is explicitly defined and disambiguated in Table 2 of the paper. For example, navigation lane change is defined as “change to the target lane as indicated by navigation,” while efficiency lane change is defined as “overtaking slow vehicles, avoiding stationary vehicles, and handling construction or congestion.” These correspond to clearly different driving intents and scene types (route-following vs. traffic-efficiency optimization).
>
>
> To prevent overlap and ambiguity during benchmark construction, we adopt a two-step process:
>
> **Logic-based filtering**: rule-based logic on navigation signals, traffic context, and vehicle behavior is used to assign and separate action categories;
>
> **Human verification**: typical cases for each action type are manually checked to ensure that samples in the benchmark are representative and consistent with the formal definitions.
>
>
> ## Response to Question 2
>
>
> Yes. As shown in Figure 2, we explicitly analyze and report the data distribution across vision, language, and action levels. The benchmark is constructed to ensure that each layer of the V-L-A hierarchy has sufficient and representative coverage, rather than being dominated by trivial actions such as going straight.

---

### Meta-Review · Area_Chair_gE1P · 2026-01-01

**Summary:**

This paper introduces DriveAction, a benchmark comprising over 16k QA pairs from real-world driving scenarios to evaluate the effectiveness of Vision-Language-Action (VLA) models. Reviewers KDFH, HC2D, MFt4, and 5d5K acknowledged authors' efforts in data collection but raised concerns about the benchmark's design and evaluation. Specifically, reviewers MFt4 and 5d5K argued that an open-loop, multiple-choice QA task does not sufficiently reflect real-time control and long-horizon decision consistency and closed-loop nature of autonomous driving tasks. They noted the lack of correlation with standard driving metrics like collision rates or displacement errors. Additionally, Reviewer HC2D mentioned the main findings of this work were intuitive and lacked novelty compared to existing works like DriveLM.

**Reviewer Concerns:**

The authors provided explanation about the annotation process and the distinction from recent works like VLM-AD, which aims at addressing Reviewer KDFH's request for transparency and related work comparison. However, the core concerns raised by MFt4 and 5d5K regarding the validity of the multiple-choice QA tasks as a proxy for driving ability evaluation. The authors argued that the benchmark serves as a diagnostic tool for VLMs prior to closed-loop integration, but they did not provide the requested evidence that connects benchmark performance to actual driving performance. Consequently, they question the benchmark's practical value on developing autonomous driving agents.

**Reviewer Scores:**

Reviewer KDFH (Score 4) might maintain their score or increase it to 5, as their specific questions were answered, but the concern about its practical impact remains. Reviewer HC2D (Score 4) likely maintained their score, as the novelty of the findings was not sufficiently explained. Reviewers MFt4 (Score 2) and 5d5K (Score 2) would likely not change their scores, as the rebuttal confirmed the lack of closed-loop evaluation or correlation with driving metrics, which is their primary reason for rejection.

---

### Decision · Program_Chairs · 2026-01-26

Reject